# Sodium butyrate modulates chicken macrophage proteins essential for *Salmonella* Enteritidis invasion

**Anamika Gupta[1]\*, Mohit Bansal[1], Rohana Liyanage[2], Abhinav Upadhyay[3], Narayan Rath[4], Annie Donoghue[4], Xiaolun Sun[1]\***

1 Department of Poultry Science, University of Arkansas, Fayetteville, Arkansas, United States of America,
2 Department of Chemistry, University of Arkansas, Fayetteville, Arkansas, United States of America,
3 Department of Animal Science, University of Connecticut, Storrs, Connecticut, United States of America,
4 Poultry Production and Product Safety Research Unit, United States Department of Agriculture-Agriculture Research Station, Fayetteville, Arkansas, United States of America

\* xiaoluns@uark.edu (XS); ag048@uark.edu (AG)

## Abstract

*Salmonella* Enteritidis is an intracellular foodborne pathogen that has developed multiple mechanisms to alter poultry intestinal physiology and infect the gut. Short chain fatty acid butyrate is derived from microbiota metabolic activities, and it maintains gut homeostasis. There is limited understanding on the interaction between *S.* Enteritidis infection, butyrate, and host intestinal response. To fill this knowledge gap, chicken macrophages (also known as HTC cells) were infected with *S.* Enteritidis, treated with sodium butyrate, and proteomic analysis was performed. A growth curve assay was conducted to determine sub-inhibitory concentration (SIC, concentration that do not affect bacterial growth compared to control) of sodium butyrate against *S.* Enteritidis. HTC cells were infected with *S.* Enteritidis in the presence and absence of SIC of sodium butyrate. The proteins were extracted and analyzed by tandem mass spectrometry. Our results showed that the SIC was 45 mM. Notably, *S.* Enteritidis-infected HTC cells upregulated macrophage proteins involved in ATP synthesis through oxidative phosphorylation such as ATP synthase subunit alpha (ATP5A1), ATP synthase subunit d, mitochondrial (ATP5PD) and cellular apoptosis such as Cytochrome-c (CYC). Furthermore, sodium butyrate influenced *S.* Enteritidis-infected HTC cells by reducing the expression of macrophage proteins mediating actin cytoskeletal rearrangements such as WD repeat-containing protein-1 (WDR1), Alpha actinin-1 (ACTN1), Vinculin (VCL) and Protein disulfide isomerase (P4HB) and intracellular *S.* Enteritidis growth and replication such as V-type proton ATPase catalytic subunit A (ATPV1A). Interestingly, sodium butyrate increased the expression of infected HTC cell protein involving in bacterial killing such as Vimentin (VIM). In conclusion, sodium butyrate modulates the expression of HTC cell proteins essential for *S.* Enteritidis invasion.

**Data Availability Statement:** All relevant data are within the paper.

**Funding:** This research was supported by grants of Arkansas Biosciences Institute, USDA National

Institute of Food and Agriculture (NIFA) Hatch project 1012366, NIFA Hatch/Multi State project 1018699, NIFA project 2020-67016-31346, and NIFA SAS 2019-69012-29905 to X. Sun. USDA grants to A. Donoghue. Poultry Federation Scholarship to A. Gupta. The funders had no role in study design, data collection and analysis, decision to publish, or preparation of the manuscript.

**Competing interests:** The authors have declared that no competing interests exist.

## Introduction

Salmonellosis is one of the globally leading food borne bacterial infectious enteritis [1,2]. *Salmonella* Enteritidis is the main pathogen and is asymptomatically colonized in the gastrointestinal tract (GIT) of its reservoir poultry [3]. The birds shed the pathogen in feces and contaminate carcass and egg yolk and shell membrane [4,5]. Consumption of contaminated and not well-cooked poultry meat, eggs and byproducts is the main cause of salmonellosis [3,6–8]. Despite various pre- and post-harvest interventions to reduce *Salmonella* Enteritidis, salmonellosis incidences remain high because the pathogen has evolved multiple adaptation strategies to evade the interventions and persistently colonize the chicken GIT [9,10].

*S.* Enteritidis has evolved strategies to alter animal cell physiology for colonizing and invading the host GIT [11,12]. *S.* Enteritidis after coming in contact with human intestinal epithelial cells secretes bacterial effector proteins such as SopE, SopE2 and SopB through *Salmonella* Pathogenicity Island (SPI-1) encoded T3SS to influence actin cytoskeleton rearrangements for its invasion of intestinal epithelial cells [11]. *S.* Enteritidis induces an innate inflammatory response, diarrhea, and systemic illness after its invasion in the intestinal epithelial cells [13,14]. *S.* Enteritidis infection induces intestinal pro-inflammatory cytokines such as *Il1β* and *Il8*, and the resulted intestinal inflammation promotes the pathogen dissemination through macrophages [15]. *S.* Enteritidis invades intestinal macrophages through micropinocytosis and enclosed in *Salmonella* containing vacuole (SCV) inside the macrophages. The *Salmonella* Pathogenicity Island (SPI-2) encoded T3SS present within the SCV and secretes effector proteins such as SseJ, SpvB, SseC for its survival and intracellular replication [16,17].

Efforts have been taken to reduce *S.* Enteritidis colonization and persistence in chicken GIT and subsequent salmonellosis but with a limited success. Microbiota metabolites, such as short chain fatty acids (SCFA), are main energy sources for colonocytes, enhance epithelial barrier integrity, and inhibit inflammation [18,19]. SCFA butyric acid has been Generally Recognised as Safe (GRAS) antimicrobials for use in foods (Butyric acid- 21CFR182.60) [20]. We have recently found that sodium butyrate effectively reduced *S.* Enteritidis attachment and invasion into the primary chicken enterocytes [21]. We also reported that sodium butyrate reduced *S.* Enteritidis invasion and inflammatory genes (*Il1β*, *Il8* and *Mmp9*) expression in chicken macrophages (HTC cells). Although those findings revealed the effect of sodium butyrate on *S.* Enteritidis infection in HTC cells at the transcriptional level, it remains poorly understood on their effect on the HTC cells at the translational level. In this study, we hypothesized that sodium butyrate modulated protein expression in HTC cells infected with *S.* Enteritidis invasion. Using a sub-inhibitory dose to *S.* Enteritidis, we found that sodium butyrate induced various protein expression in HTC cells and the protein changes were related to host response to *S.* Enteritidis invasion and survival. The findings from this study will help the development of new intervention against *S.* Enteritidis infection.

## Materials and methods

### Chicken macrophage cell line

A naturally transformed line of chicken macrophages named HTC cells [22] were cultured in Roswell Park Memorial Institute (RPMI) 1640 media (Thermo Fisher Scientific, Carlsbad, CA) containing 10% fetal bovine serum (Thermo Fisher Scientific), 1X antibiotic antimycotic solution (Sigma-Aldrich, St Louis, MO, USA), 1X sodium pyruvate solution (Sigma-Aldrich), gentamicin solution (Sigma-Aldrich), 10 mM glutamine solution (Thermo Fisher Scientific) at 37˚C for 24–48 h in a humidified incubator containing 5% $CO_2$ as described earlier with minor modifications. The cells were cultured to semi-confluence followed by dissociation with Accumax (Sigma-Aldrich) to perform different assays.

## Bacterial strain and culture condition

*S.* Enteritidis GFP 338 was cultured in 10 mL of tryptic soy broth (TSB; Hardy Diagnostics CRITERION™, Santa Maria, CA, USA) at 37°C for 18 h. Following subculture in 10 mL TSB for additional 10 h, the culture was centrifuged at 4000 rpm for 10 min. The pellet was suspended in sterilized phosphate buffer saline (PBS, pH 7) and used as the inoculum. The enumeration of *S.* Enteritidis counts in inoculum was made by plating serial 5-fold dilutions on brilliant green agar (BGA; Difco Laboratories, Detroit, Michigan, USA) and the plates were incubated at 37°C for 24 h for bacterial enumeration.

## Determination of SIC of sodium butyrate

SIC of sodium butyrate against *S.* Enteritidis was determined according to a previous published procedure [23,24] with minor modifications. Sterile 96-well polystyrene tissue culture plate (Costar, Corning Incorporated, Corning, NY) containing twofold dilutions of sodium butyrate (363, 181.5, 90.75, 45, 22 and 11 mM) in TSB was inoculated with ~6.0 Log CFU of *S.* Enteritidis along with a negative control (no butyrate) and the plate was incubated at 37°C for 24 h under aerobic condition. The highest concentration of sodium butyrate that did not inhibit *S.* Enteritidis growth after 24 h of incubation was determined as the SIC for the present study. The growth of *S.* Enteritidis was determined by measuring absorbance using spectrophotometric microplate reader (Benchmark; Bio-Rad Laboratories, Hercules, CA, USA) at 570 nm.

## Effect of SIC of sodium butyrate on cell viability of HTC cells

Based on the determination of SIC, its effect on viability of HTC cells in response to sodium butyrate was determined by 3-[4,5-dimethylthiazole-2-yl]-2,5-diphenyltetrazolium bromide (MTT) assay [25,26]. HTC cells ($10^4$ cells/well) were seeded in a 96-well plate for 48 h at 37°C in a humidified incubator containing 5% $CO_2$ to form a monolayer. The HTC cells were incubated with SIC of sodium butyrate for 4 h at 37°C. The MTT reagent (10 μL) was added to HTC cells and incubated at 37°C for 2 h. After removing the supernatant, 100 μL isopropanol (Sigma-Aldrich) was added and the plate was incubated at room temperature in dark for 1 h. The absorbance was measured at 570 nm by using spectrophotometric microplate reader (Benchmark; Bio-Rad Laboratories, Hercules, CA, USA).

## Proteomic sample preparation and in-gel protein digestion

HTC cells ($10^5$ cells per well) were seeded into 6-well plate (Costar) in RPMI 1640 media containing 10% FBS and incubated for 48 h at 37°C in a humidified, 5% $CO_2$ incubator to form a monolayer. A mid-log phase (10 h) culture of *S.* Enteritidis was inoculated on HTC cells ($\sim$6 Log CFU/mL; multiplicity of infection 10:1) in the presence or absence of 45 mM sodium butyrate. Infected HTC cells were incubated for 4 h followed by rinsing with serum free RPMI 1640 media twice. HTC cells were then lysed by M-PER™ Mammalian Protein Extraction Reagent (Thermo Fisher Scientific) as described earlier [27,28].

Cell lysate were subjected to 4–20% gradient SDS Page gel electrophoresis and each sample was run in triplicate. Gel was stained with Coomassie blue and the gel segments were excised and triturated into small pieces followed by washing with 25 mM ammonium bicarbonate ($NH_4HCO_3$, Thermo Fisher Scientific). Destaining of gel segments was performed by adding 50% Acetonitrile (ACN, Bio-Rad Laboratories, Hercules, CA, USA) in 25 mM $NH_4HCO_3$ for 1 h followed by decanting all the detaining solution. Subsequently, 100% ACN was added to dehydrate gel pieces and evaporated to the dryness using Labconco Centriyap. Reduction of

proteins was performed by adding 10 mM dithiothreitol (DTT, Bio-Rad) in 25 mM NH$_4$HCO$_3$ (1.5 mg/mL) to the dried gel pieces and by keeping it at 60˚C for 1 h. After 1 h, excess DTT was discarded and proceeded to alkylation 55 mM iodoacetamide (Bio-Rad) with 25 mM NH$_4$HCO$_3$ (10 mg/mL) at room temperature for 1 h in the dark. Excess iodoacetamide was completely removed and the gel pieces were rinsed with 25 mM NH$_4$HCO$_3$ followed by dehydration of gel pieces with ACN. Dehydrated gel pieces were then vacuum-dried before adding MS Grade Trypsin (20 ng/mL in 25 mM NH$_4$HCO$_3$) and incubated overnight at 37˚C. The extracted peptides were dried completely and resuspended in 0.1% formic acid for analyses by Liquid chromatography tandem mass spectrometry (LC-MS/MS). Three samples were used for each group and data were analyzed individually for each sample.

**Mass spectrometry analysis.** LC-MS/MS was performed using an Agilent 1200 series micro-flow high-performance liquid chromatography (HPLC) coupled to a Bruker AmaZon SL quadrupole ion trap mass spectrometer (Bruker Daltoniks Inc., Billerica, MA, United States) with a captive spray ionization source as described earlier [27–29]. Tryptic peptides were separated by using C$_{18}$ capillary column (150 mm × 0.1 mm, 3.5 μm particle size, 300 Å pore size; ZORBAX SB) with 5–40% gradients of 0.1% formic acid (solvent A) and ACN in 0.1% formic acid (solvent B). Solvent flowed at a rate of 4 μL/min over a duration of 300 min each.

LC-MS/MS data were acquired in positive ion mode. Bruker captive electro spray source was operated with a dry gas temperature of 150˚C and a dry nitrogen flow rate of 3 L/min with captive spray voltage of 1500 volts. The data acquisition was in the Auto MS (n) mode optimized the trapping condition for the ions at m/z 1000. MS scans were performed in enhanced scanning mode (8100 m/z/second), MS/MS fragmentation scans performed automatically for top 10 precursor ions. The samples were run three times for each group as technical replicates and experiment was repeated two times for analyzing results.

By using Bruker Data Analysis 4.0 software, peaks were picked from LC-MS/MS chromatogram using default peak picking method recommended and to created Protein Analysis Results.xml file. This was used for searching Mascot database. In Mascot search, parent ion and fragment ion mass tolerances were set at 0.6 Da with cysteine carbamidomethylation as fixed modification and methionine oxidation as variable modifications. For the identification of proteins in cell extracts, Mascot search was performed against Gallus UniProt database. Identification of proteins is with 95% confidence limit and with less than 5% false discovery rate (FDR). FDR was calculated in during the Mascot search by simultaneously searching the reverse sequence database. Uncharacterized Gallus proteins were identified based on gene sequence similarities tentatively. For evaluation of differentially expressed proteins, Mascot.dat files_-were exported to Scaffold Proteome Software version 4.8 and quantitative differences were determined based on 95% confidence limit. To determine the signaling pathway of proteins, the differentially regulated proteins were analyzed using software such as Protein Analysis through Evolutionary Relationships software (PANTHER) and STRING protein association network (FDR 0.05) as described as before [28].

## Statistical analysis

The CFU counts of *S*. Enteritidis were logarithmically transformed (Log CFU) to maintain homogeneity of variance [30]. In the present study, we used triplicate samples and the experiment was repeated twice. Cell viability data was analyzed by using t-test in Graph-pad 7 Software. Scaffold Proteome Software version 4.8 (Proteome Software Inc, Portland, OR) was used to analyze Mascot files for the proteomic analysis. Differentially expresses proteins were determined using Student's t-test and probability of $P<0.05$ was required for statistically significant differences.

## Results

### SIC of sodium butyrate against *S*. Enteritidis

The SIC of sodium butyrate against *S*. Enteritidis was determined based on growth curve analysis. The three concentrations of sodium butyrate that did not reduce *S*. Enteritidis growth after 12 h of incubation at 37˚C were 11, 22 and 45 mM [21]. Therefore, we have selected the highest SIC 45 mM of sodium butyrate to culture HTC cells and study the global protein expression by proteomic assay. Next, to assess this SIC impacted HTC cells growth, the cell was culture in the presence of 45 mM butyrate. Sodium butyrate at the SIC did not reduce the growth of *S*. Enteritidis compared to the control HTC cells (*P*>0.05).

### Effect of *S*. Enteritidis on the proteome of HTC cells

Next, to reveal the translation alterations, HTC cells were cultured with *S*. Enteritidis. The proteins were extracted and assessed by tandem mass spectrometry, and the data were analyzed. A total of 389 proteins were identified when HTC cells were infected with *S*. Enteritidis. Quantitative comparison showed that *S*. Enteritidis infection downregulated 22 proteins and upregulated 9 proteins compared to uninfected HTC cells (*P*<0.05), however 358 proteins were not affected (*P*>0.05).

Specifically, *S*. Enteritidis infection in HTC cells downregulated the protein expression correlated with various biological process. Particularly, proteins related with biological regulation such as Non-specific serine/threonine protein kinase (ATM), Anaphase promoting complex subunit 1 (ANAPC1), Zinc finger protein 462 (E1C5J4), Actin-related protein 3 (ACTR3) and Zinc finger homeobox protein 4 (ZFHX4) were downregulated by *S*. Enteritidis infection in HTC cells. In addition, proteins related with cellular component biogenesis such as ATM, ANAPC1, Centromere protein E (CENPE), Hsc70-interacting protein (ST13), E1C5J4 and ACTR3; cellular process such as ATM, ANAPC1, CENPE, Natural killer cell triggering receptor (NKTR), Ubiquitin-conjugating enzyme E2 (UBE2K), ST13, E1C5J4, ACTR3 and ZFHX4 were also downregulated after *S*. Enteritidis infection in HTC cells. Likewise, proteins involved in localization such as CENPE; metabolic process such as ATM, ANAPC1, UBE2K, E1C5J4 and ZFHX4; developmental and multicellular organismal process such as Neuron navigator-3 (NAV3); response to stimulus and signaling such as ATM were also modulated.

In contrast, *S*. Enteritidis infected HTC cells upregulated the expression of proteins correlated with distinct biological processes. *S*. Enteritidis upregulated the protein expression related with biological regulation such as Cytochrome C (CYC); cellular component biogenesis and response to stimulus includes HSPA8. In addition, *S*. Enteritidis infection in HTC cells also upregulated proteins associated with cellular process such as Heat shock cognate 71 kDa protein (HSPA8), ATP synthase subunit-d (ATP5PD), Peptidylprolyl isomerase (FKBP12), CYC, Bifunctional purine biosynthesis protein (ATIC) and Hydroxymethylbilane synthase (HMBS); localization such as CYC, HSPA8 and ATP5PD and metabolic process such as HSPA8, ATP5PD, FKBP12, CYC, ATIC and HMBS (Tables 1 and 2, Fig 1).

Moreover, signaling pathway analysis by STRING predicted that that *S*. Enteritidis infection in HTC cells downregulated proteins involved in nucleotide binding, cytoplasmic and cytoskeletal changes, and actin binding (Table 3). Additionally, *S*. Enteritidis infected HTC cells upregulated proteins related with various metabolic pathways (Table 4).

### Effect of sodium butyrate on the proteome of HTC cells infected with *S*. Enteritidis

HTC cells infected with *S*. Enteritidis in the presence of sodium butyrate downregulated 14 proteins and upregulated 6 proteins compared to HTC cells infected with *S*. Enteritidis alone

**Table 1. Differentially regulated proteins in HTC cells after *S*. Enteritidis infection.**

| Proteins (Downregulated proteins) | Alternate ID by Gene | UNIPROT Accession number | Molecular Weight | Fold change by category (SE/Control) | t-TEST (P-VALUE) P<0.05 |
|---|---|---|---|---|---|
| Uncharacterized protein | DNAH9 | F1NVK1 | 482 | 0.2 | 0.032 |
| Ryanodine receptor 2 | | F1NLZ9 | 563 | 0.2 | 0.035 |
| Uncharacterized protein | NAV3 | F1NAH8 | 250 | 0 | 0.0082 |
| Biorientation of chromosomes in cell division 1 like 1 | BOD1L1 | R4GKR8 | 329 | 0 | 0.032 |
| Actin-related protein 3 | ACTR3 | ARP3 | 47 | 0.4 | 0.0074 |
| Zinc finger homeobox protein 4 | ZFHX4 | ZFHX4 | 395 | 0 | 0.0049 |
| Non-specific serine/threonine protein kinase | ATM | E1C0Q6 | 348 | 0.2 | 0.022 |
| Spectrin beta chain | SPTBN1 | A0A1D5PJY1 | 274 | 0 | 0.013 |
| Collagen type V alpha 2 chain | COL5A2 | A0A1D5P6W1 | 145 | 0.1 | 0.029 |
| Elongation factor 1-alpha | EEF1A1 | A0A1L1RRR1 | 49 | 0.2 | 0.02 |
| Zinc finger protein 462 | | E1C5J4 | 278 | 0 | 0.014 |
| Actin-related protein 2/3 complex subunit 4 | ARPC4 | F1P010 | 20 | 0.3 | 0.049 |
| Anaphase promoting complex subunit 1 | ANAPC1 | E1C2U7 | 216 | 0 | 0.024 |
| Uncharacterized protein | CENPE | E1BQJ6 | 258 | 0 | 0.001 |
| Uncharacterized protein | GMFB | A0A1D5PTE8 | 17 | 0.09 | 0.015 |
| Uncharacterized protein | A0A1D5P0W7 | A0A1D5P7P7 | 25 | 0.2 | 0.02 |
| Natural killer cell triggering receptor | NKTR | A0A1D5PRM6 | 161 | 0 | 0.0069 |
| Hsc70-interacting protein | ST13 | A0A1L1RVN1 | 30 | 0 | 0.029 |
| Terpene cyclase/mutase family member | LSS | A0A1D5PDR0 | 85 | 0 | 0.032 |
| Adseverin OS = Gallus gallus | SCIN | A0A1D5PBC3 | 79 | 0.1 | 0.011 |
| Uncharacterized protein | UBE2K | A0A1L1RJI2 | 22 | 0.1 | 0.0087 |
| Histidine triad nucleotide binding protein 2 | HINT2Z | R4GGS3 | 17 | 0 | 0.0076 |
| Heat shock cognate 71 kDa protein | HSPA8 | F1NWP3 | 71 | 1.2 | 0.0017 |
| Bifunctional purine biosynthesis protein | ATIC | F7AXZ3 | 69 | 1.3 | 0.025 |
| Peptidylprolyl isomerase | FKBP12 | Q90ZG0 | 12 | 2 | 0.043 |
| Hydroxymethylbilane synthase | HMBS | A0A1D5NYN8 | 37 | 2.6 | 0.04 |
| EF-hand domain family member D2 | EFHD2 | A0A1D5PD25 | 25 | 3.5 | 0.017 |
| Uncharacterized protein | | A0A1D5P4K6 | 20 | 2.5 | 0.02 |
| ATP synthase subunit d, mitochondrial | ATP5PD | E1C658 | 18 | 2 | 0.042 |
| ATP synthase subunit alpha | ATP5A1 | A0A182C637 | 60 | 1.3 | 0.039 |
| Cytochrome c | CYC | CYC | 12 | 12 | 0.03 |

(*P*<0.05), whereas 369 proteins were not affected (*P*>0.05). Specifically, sodium butyrate treatment in *S*. Enteritidis infected HTC cells downregulated proteins allied with different biological processes for example biological regulation such as WD repeat-containing protein-1 (WDR1) and cellular component biogenesis such as ATP-dependent 6-phosphofructokinase (PFKP) and WDR1. Similarly, proteins associated with cellular process such as Protein disulfide-isomerase (P4HB), WDR1, PFKP and Rab GDP dissociation inhibitor (F1NCZ2); localization such as F1NCZ2; metabolic process such as PFKP and response to stimulus such as P4HB was modulated by sodium butyrate treatment in *S*. Enteritidis infected macrophages.

In contrast, sodium butyrate treatment in *S*. Enteritidis infected HTC cells upregulated proteins related with cellular component biogenesis such as HSPB9, Ras-related protein Rab-11A

**Table 2. Go-annotated proteins associated with different biological processes in HTC infected with *S.* Enteritidis.**

| Functional Annotations | Downregulated Proteins | Upregulated Proteins |
|---|---|---|
| Biological regulation | Non-specific serine/threonine protein kinase (ATM), Anaphase promoting complex subunit 1 (ANAPC1), Zinc finger protein 462 (E1C5J4), Actin-related protein 3 (ACTR3) and Zinc finger homeobox protein 4 (ZFHX4) | Cytochrome C (CYC) |
| Cellular component biogenesis | ATM, ANAPC1, Centromere protein E (CENPE), Hsc70-interacting protein (ST13), E1C5J4 and ACTR3 | Heat shock cognate 71 kDa protein (HSPA8) |
| Cellular process | ATM, ANAPC1, CENPE, Natural killer cell triggering receptor (NKTR), Ubiquitin-conjugating enzyme E2 (UBE2K), ST13, E1C5J4, ACTR3 and ZFHX4 | HSPA8, ATP synthase subunit-d (ATP5PD), Peptidylprolyl isomerase (FKBP12), CYC, Bifunctional purine biosynthesis protein (ATIC) and Hydroxymethylbilane synthase (HMBS) |
| Localization | CENPE | CYC, HSPA8 and ATP5PD |
| Metabolic process | ATM, ANAPC1, UBE2K, E1C5J4 and ZFHX4 | HSPA8, ATP5PD, FKBP12, CYC, ATIC and HMBS |
| Developmental process | Neuron navigator-3 (NAV3) | – |
| Multicellular organismal process | Neuron navigator-3 (NAV3) | – |
| Response to stimulus | ATM | HSPA8 |
| Signaling | ATM | – |

(RAB11A), Vimentin (VIM) and Actin-related protein 2/3 complex (ARPC4); cellular process such as RAB11A, VIM, ATPSF1B and ARPC4; and metabolic process such as ENO1 and ATP5F1B (Tables 5 and 6, Fig 2).

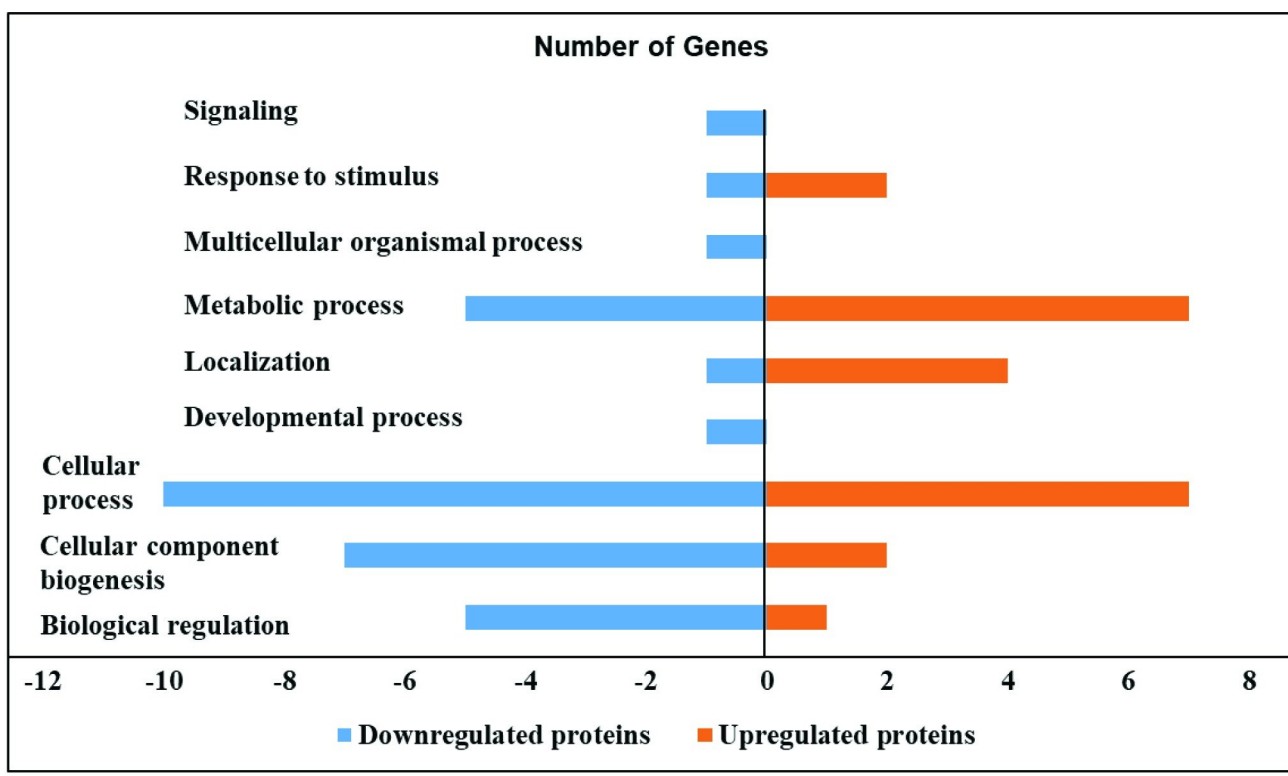

**Fig 1. Effect of *S.* Enteritidis on the proteome of HTC cells.** *S.* Enteritidis infection in HTC cells induced down and upregulated proteins in different biological processes. HTC cells were treated with *S.* Enteritidis for 4 h, proteins were extracted and analyzed by tandem mass spectrometry. Differentially expressed proteins were calculated using Scaffold software (*P*<0.05) and biological processes were predicted by using STRING and PANTHER software.

**Table 3. Pathways downregulated by *S*. Enteritidis infected HTC cells.**

| S. No | Pathway ID | Pathway description | Count in gene set | False discovery rate |
|-------|-----------|---------------------|-------------------|----------------------|
| 1 | GO0051303 | Establishment of chromosome localization | 2 | 0.0337 |
| 2 | GO0030833 | Regulation of actin filament polymerization | 2 | 0.0337 |
| 3 | GO0007049 | Cell cycle | 3 | 0.0337 |
| 4 | GO0043232 | Intracellular non-membrane-bounded organelle | 5 | 0.009 |
| 5 | KW0206 | Cytoskeleton | 5 | 0.00028 |
| 6 | KW0009 | Cytoplasm | 7 | 0.0018 |
| 7 | KW0547 | Nucleotide binding | 6 | 0.0167 |
| 8 | KW0009 | Actin-binding | 4 | 0.00047 |

Signaling pathway analysis by STRING predicted that sodium butyrate downregulated the protein expression of *S*. Enteritidis infected HTC cells plays role in actin filament binding, intracellular membrane bound changes, cellular homeostasis, and cortical actin cytoskeletal changes (Table 7). Additionally, sodium butyrate upregulated the proteins involved in bacterial killing, membrane trafficking and cytoskeleton changes.

## Discussion

*S*. Enteritidis is an intracellular pathogen and induces an inflammatory immune response in GIT to overwhelm commensal microbiota, colonize and invade the intestinal cells [31–33]. The pathogen invades into chicken intestinal macrophages and disseminates systemically [11,16,34,35]. However, it remains largely elusive the mechanism of *S*. Enteritidis invading chicken intestine at the cellular and molecular level. In this study, we have investigated the effect of sodium butyrate on the proteomics of macrophage HTC cells infected with *S*. Enteritidis. We found that various proteins in the HTC cells were modulated by *S*. Enteritidis infection and sodium butyrate.

Notably, *S*. Enteritidis infection downregulated the expression of macrophage cellular proteins that regulate actin cytoskeletal rearrangements such as SCIN, ACTR3 and ARPC4 as compared to uninfected cells. *S*. Enteritidis has evolved many strategies to manipulate host actin cytoskeletal rearrangements for its internalization [9]. *S*. Enteritidis invades intestinal epithelium through an array of bacterial effector proteins using type III secretion system (T3SS) [36]. After invasion of *S*. Enteritidis in the epithelial cells, there is reorganization of actin cytoskeletal by constitution of microvilli, recession of membrane ruffling and restoration of epithelium through actin binding proteins [37–41]. Our results indicate that *S*. Enteritidis infection of the HTC cells downregulated proteins related with reorganization of actin cytoskeleton, possibly facilitating its endocytosis inside the macrophages.

Interestingly, *S*. Enteritidis infection upregulated HTC cellular proteins that maintain ATP synthesis such as ATP5A1 and ATP5PD. ATP synthase proteins are crucial for maintenance of cellular homeostasis and cell energy metabolism through oxidative phosphorylation via ATP synthesis [42–44]. The synthesized ATP could provide vital energy source for the pathogen

**Table 4. Pathways upregulated by *S*. Enteritidis infected HTC cells.**

| S. No | Pathway ID | Pathway description | Count in gene set | False discovery rate |
|-------|-----------|---------------------|-------------------|----------------------|
| 1 | GGA1592230 | Mitochondrial biogenesis | 2 | 0.0016 |
| 2 | GGA163200 | Respiratory electron transport, ATP synthesis | 2 | 0.007 |
| 3 | GGA01100 | Metabolic pathways | 4 | 0.0084 |
| 4 | GO0009167 | Purine ribonucleoside monophosphate metabolic process | 2 | 0.0197 |

**Table 5. Differentially regulated proteins by sodium butyrate treatment in *S*. Enteritidis infected HTC cells.**

| Proteins (Downregulated proteins) | Alternate ID by Gene | UNIPROT Accession number | Molecular Weight | Fold change by category (SB +SE/SE) | t-TEST (P-VALUE) $P<0.05$ |
|---|---|---|---|---|---|
| Alpha-actinin-1 | ACTN1 | A0A1D5P9P3 | 102 | 0.6 | 0.0084 |
| Protein disulfide-isomerase | P4HB | PDIA1 | 57 | 0.6 | 0.038 |
| Rab GDP dissociation inhibitor | GDI2 | F1NCZ2 | 51 | 0.7 | 0.043 |
| ATP-dependent 6-phosphofructokinase | PFKP | A0A1D5P0Z0 | 86 | 0.2 | 0.037 |
| Vinculin | VCL | VINC | 125 | 0.4 | 0.014 |
| Uncharacterized protein | RCJMB04_4k19 | Q5ZLW0 | 70 | 0.3 | 0.021 |
| V-type proton ATPase catalytic subunit A | ATP6V1A | F1NBW2 | 68 | 0.2 | 0.017 |
| Ubiquitin carboxyl-terminal hydrolase | UCHL3 | F1NY51 | 22 | 0.2 | 0.03 |
| Cathepsin D | CTSD | CATD | 43 | 0.2 | 0.0092 |
| NADPH—cytochrome P450 reductase | POR | F1P2T2 | 77 | 0.3 | 0.041 |
| Uncharacterized protein | IDI1 | F1NZX3 | 33 | 0.1 | 0.0041 |
| WD repeat-containing protein 1 | WDR1 | F1NRI3 | 67 | 0.09 | 0.041 |
| EF-hand domain family member D2 | EFHD2 | A0A1D5PD25 | 25 | 0.2 | 0.0066 |
| Pyridoxal phosphate homeostasis protein OS | PROSC | E1C516 | 30 | 0 | 0.04 |
| Alpha-enolase | ENO1 | A0A1L1RKH8 | 49 | 1.3 | 0.048 |
| ATP synthase subunit beta, mitochondrial | ATP5F1B | ATPB | 57 | 1.3 | 0.028 |
| Ras-related protein Rab-11A | RAB11A | RB11A | 24 | 2.2 | 0.03 |
| Uncharacterized protein | HSPB9 | A0A1L1RXQ8 | 21 | 2.4 | 0.013 |
| Actin-related protein 2/3 complex subunit 4 | ARPC4 | F1P010 | 20 | 4.4 | 0.0023 |
| Vimentin | VIM | VIME | 53 | 5.6 | 0.03 |

survival and growth. In this study, cytochrome C protein of CYC in the HTC cells was upregulated by *S*. Enteritidis infection. During cellular apoptosis, cytochrome C is released in cytoplasm from the permeabilization of mitochondrial outer membrane, which activates apoptosis-promoting proteins such as apoptotic protease activating factor-1 (Apaf-1) [45,46]. Our results showed that *S*. Enteritidis infection upregulates proteins associated with ATP synthesis and cell apoptosis.

In *S*. Enteritidis-infected HTC cells, sodium butyrate downregulated proteins associated with disassembly of actin filament and stimulation of actin polymerization and binding such as WDR1, VCL, ACTN1, and P4HB. *S*. Enteritidis colonization of chicken intestinal epithelial

**Table 6. Go-annotated proteins associated with different biological processes after sodium butyrate treatment in HTC cells infected with *S*. Enteritidis.**

| Functional Annotations | Downregulated Proteins | Upregulated Proteins |
|---|---|---|
| Biological regulation | WD repeat-containing protein-1 (WDR1) | – |
| Cellular component biogenesis | ATP-dependent 6-phosphofructokinase (PFKP) and WDR1 | HSPB9, Ras-related protein Rab-11A (RAB11A), Vimentin (VIM) and Actin-related protein 2/3 complex (ARPC4) |
| Cellular process | Protein disulfide-isomerase (P4HB), WDR1, PFKP and Rab GDP dissociation inhibitor (F1NCZ2) | RAB11A, VIM, ATPSF1B and ARPC4 |
| Localization | F1NCZ2 | – |
| Metabolic process | PFKP | ENO1 and ATP5F1B |
| Response to stimulus | P4HB | – |

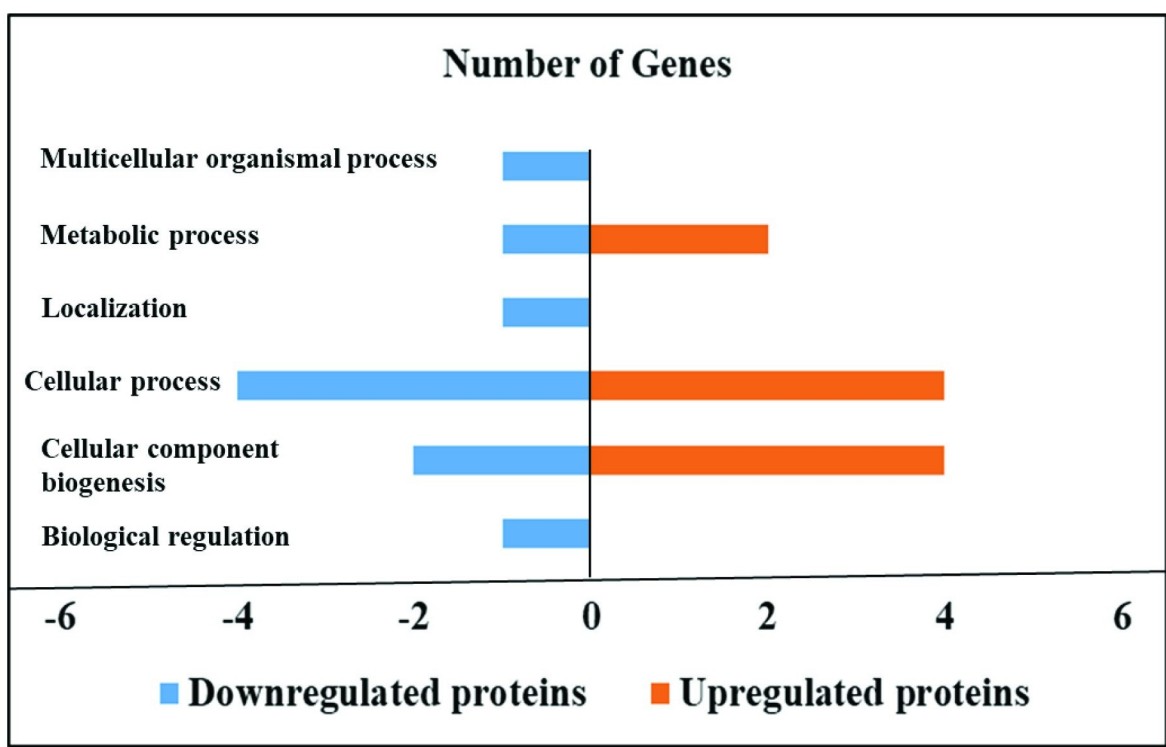

**Fig 2. Effect of sodium butyrate on the proteome of HTC cells infected with *S*. Enteritidis.** Sodium butyrate treatment in *S*. Enteritidis infected HTC cells induced down and upregulated proteins in different biological processes. HTC cells were treated with *S*. Enteritidis for 4 h in the presence and absence of sodium butyrate, proteins were extracted and analyzed by tandem mass spectrometry. Differentially expressed proteins were calculated using Scaffold software (*P*<0.05) and biological processes were predicted by using STRING and PANTHER software.

cells alter cellular functions such as cytoskeletal architecture, signal transduction and cell migration for its invasion [47]. WDR1 is an actin interacting protein responsible for actin filament dynamics and cytoskeleton regulation [48,49]. VCL is a cytoskeletal actin binding protein and maintains various physiological processes, such as adhesion and motility by promoting actin polymerization and binding to specific phospholipids [50,51]. ACTN1 is a cytoskeleton actin binding protein and regulates cell-cell matrix adhesion and cell migration [52]. In addition, sodium butyrate also downregulated a vacuolar ATPase proton pump protein ATP6V1A that acidifies intracellular compartments to increase permeability of endosomes, and results in vesicular swelling, and intracellular bacterial growth [53]. Expression of ATPV1A in macrophages is increased during *Salmonella* infection and intracellular replication [53,54]. Together, the reduction of these proteins by sodium butyrate might decrease *S*. Enteritidis invasion in the HTC cells.

**Table 7. Pathways downregulated by sodium butyrate treatment in *S*. Enteritidis infected HTC cells.**

| S. No | Pathway ID | Pathway description | Count in gene set | False discovery rate |
|---|---|---|---|---|
| 1 | GO0019725 | Cellular homeostasis | 2 | 0.0185 |
| 2 | GO0009653 | Anatomical structure morphogenesis | 3 | 0.0185 |
| 3 | GO0043231 | Intracellular membrane-bounded organelle | 4 | 0.0209 |
| 4 | GO0051015 | Actin filament binding | 3 | 0.00019 |
| 5 | GO0017166 | Vinculin binding | 2 | 0.00025 |
| 6 | GO0030864 | Cortical actin cytoskeleton | 3 | 1.70E-05 |

Interestingly, sodium butyrate upregulated HTC cell cytoskeleton protein VIM that maintains cell integrity and many cellular processes such as cell adhesion, immune response, and autophagy [55]. VIM released by activated macrophages promotes production of oxidative metabolites and bacterial killing in response to pro-inflammatory signaling pathways [56]. *Salmonella* infection in chicken macrophages promotes pro-inflammatory cytokine immune response for its invasion and survival [21]. It is necessary to investigate whether sodium butyrate-upregulated VIM protein activates inflammatory response.

## Conclusion

This study showed that butyrate reduced the cellular actin and cytoskeleton rearrangement proteins in *S*. Enteritidis infected HTC cells. In addition, sodium butyrate upregulated proteins enhancing pro-inflammatory response in *S*. Enteritidis infected HTC cells. Collectively, these results suggest that sodium butyrate modulates HTC cell protein expression essential for *S*. Enteritidis invasion in the chicken macrophages.

## Acknowledgments

The authors would also like to thank Scott Zornes and Sonia Tsai for their technical assistance.

## Author Contributions

**Conceptualization:** Anamika Gupta, Abhinav Upadhyay, Narayan Rath, Xiaolun Sun.

**Data curation:** Anamika Gupta, Mohit Bansal, Rohana Liyanage, Xiaolun Sun.

**Formal analysis:** Anamika Gupta, Annie Donoghue, Xiaolun Sun.

**Funding acquisition:** Annie Donoghue, Xiaolun Sun.

**Investigation:** Anamika Gupta, Mohit Bansal, Narayan Rath, Xiaolun Sun.

**Methodology:** Anamika Gupta, Rohana Liyanage, Narayan Rath, Xiaolun Sun.

**Project administration:** Anamika Gupta.

**Supervision:** Narayan Rath, Annie Donoghue, Xiaolun Sun.

**Writing – original draft:** Anamika Gupta, Mohit Bansal, Rohana Liyanage, Abhinav Upadhyay, Xiaolun Sun.

**Writing – review & editing:** Anamika Gupta, Mohit Bansal, Abhinav Upadhyay, Narayan Rath, Annie Donoghue, Xiaolun Sun.

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
