## [Decision Letter · Decision Letter 0]

23 Feb 2021

PONE-D-20-40856

Sodium butyrate modulates proteins essential for Salmonella Enteritidis invasion in chicken macrophages

PLOS ONE

Dear Dr. Sun,

Thank you for submitting your manuscript to PLOS ONE. After careful consideration, we feel that it has merit but does not fully meet PLOS ONE’s publication criteria as it currently stands. Therefore, we invite you to submit a revised version of the manuscript that addresses the points raised during the review process.

Both revised have advised that the authors pay articular attention to revising the Introduction and especial the Discussion.  Both reviewers believe that the Introduction needs be more focused on the topic at hand.  Further the Discussion requires substantial revision focusing on the data gathered from the experiments and less on speculation.  Please detail the revisions made in the submission letter with the revised manuscript.  These revisions will be the basis of acceptance or rejection of the revised manuscript.

Please submit yosur revised manuscript by Mar 29 2021 11:59PM. If you will need more time than this to complete your revisions, please reply to this message or contact the journal office at plosone@plos.org. Please include the following items when submitting your revised manuscript:

We look forward to receiving your revised manuscript.

Kind regards,

Michael H. Kogut, Ph.D.

Academic Editor

PLOS ONE

Journal Requirements:

Reviewers' comments:

Reviewer's Responses to Questions

**Comments to the Author**

1. Is the manuscript technically sound, and do the data support the conclusions?

Reviewer #1: Yes

Reviewer #2: Partly

2. Has the statistical analysis been performed appropriately and rigorously? 

Reviewer #1: Yes

Reviewer #2: Yes

3. Have the authors made all data underlying the findings in their manuscript fully available?

Reviewer #1: Yes

Reviewer #2: No

4. Is the manuscript presented in an intelligible fashion and written in standard English?

Reviewer #1: Yes

Reviewer #2: Yes

5. Review Comments to the Author

Reviewer #1: The manuscript is of great importance for Salmonella research. I am concerned about few points:

Abstract – add the definition of SIC at the first time to mentioned it.

Line 30: sub-inhibitory concentration; and Line 32: subinhibitory concentration – please, decide which one is correct the keep consistency.

Introduction: Lines 62-64: the sentence does not make sense – please, re-write

Line 65: delete “its”

Line 71: “which contributes to intestinal inflammation”

Line 76 and others: “short-chain fatty acid”

Line 77: use GIT for gastrointestinal tract after the first time

Line 86: “..level, it remains..”

What is the hypothesis of the study?

The introduction needs a revision – sometimes the sentences are not connected and lack clarity.

Discussion: Line 270: “colonize the intestine of the host”

Reviewer #2: The study reported in this manuscript uses a macrophage cell culture model with infection by Salmonella Enteritidis and treatment with a sub-inhibitory concentration (SIC) of sodium butyrate to characterize the effects of infection and of butyrate on protein levels in the cells. The design of the experiment and the wet-lab aspects of the study are generally clearly described and appear to be appropriately conducted. The presentation of results and, especially, the conclusions drawn from the data need to be re-examined and revised.

In the abstract, state what type of cells HTC are.

Because this experimental model uses a SIC of butyrate, and has verified that the butyrate does not affect bacterial growth, the logical conclusion to be drawn about any detected protein differences between infected cells with and without added butyrate should be that these protein differences are not associated with bacterial growth. In the abstract and elsewhere, some of the stated conclusions seem to be in opposition to the data generated in this study, and instead are a reiteration of conclusions of previous studies reported in the literature, which are contradicted by the current study’s findings. The authors should state conclusions that are defensible from the data of their own study and, where needed, discuss the potential reasons for discrepancies in their own data and studies reported previously by others.

Line 261. Should be reworded from “Following signaling pathway analysis by STRING revealed that…” to “Signaling pathway analysis by STRING predicted that…” The authors should be aware of the limitations of the analysis, and that it predicts but does not prove pathway involvement. And the word “Following” was not needed.

Line 304 states that S. Enteritidis leads to apoptosis, but this study does not show a change in cell survival between infected and non-infected cells. Clarify. Do the authors mean that the amount of protein(s) typically associated with apoptosis are significantly different between infected and non-infected cells?

Line 336 states enhanced bacterial killing, but this experimental model showed no effect on bacterial growth.

Lines 342-343 state that butyrate upregulates proteins that enhanced bacterial killing in HTC, however, the experimental model used shows no effect of butyrate on bacterial growth. Thus, the logical conclusion to be drawn from their own data is that these upregulated proteins are not associated with bacterial growth and therefore, at least in HTC, are not enhancing bacterial killing.

Conclusions. Two of the major conclusions are about the effect of butyrate on “cellular invasion”, however, this study is not designed to, and does not, measure cellular invasion. The stated conclusions should be supported by the data of this study. The study has an appropriate design to detect effects of bacterial infection and of butyrate; the stated conclusions should related more closely to the data of this study.

Tables 1 and 4. I cannot discern any organization of this table, because the proteins do not appear to be listed in order of any of the column headings. This makes it difficult to read. Please select a logical heading, for example the P value, and organize the table based upon it. That would add value for the reader.

Tables 2 and 3. Protein counts as low as 2 in pathways may not be highly reliable indicators of the involvement of that pathway (especially if it’s a pathway with many proteins) despite software predictions. Suggest that the authors consider more stringent thresholds for the pathways displayed so that the attention can focus on those with higher reliability. For example, an FDR of 0.02.

Figure 1. This can be deleted and stated in a single sentence in the results section.

Figure 2 legend. Please reword to state that “….biological processes were predicted..." by using STRING and PANTHER software.” The authors did not study the processes using software, they used the software to predict the involved biological processes. It’s important to recognize what the software does and does not do. It predicts (hypothesizes) the relationship of the proteins with processes.

Figure 2 and 5. In comparing these two figures, it is apparent that Figure 2 allows the inclusion of the same proteins in multiple biological pathways because the results section reported 31 proteins as up- or down-regulated but Figure 2 displays well over 50 proteins in total count. However, in Figure 5, there is an exact agreement (n = 20) between the number reported in the results section and the number displayed in the graph. Although not impossible, this is highly unlikely to have such a discrepancy in the multiple use of proteins in Figure 2 and none in Figure 5. Authors should check that they have not accidentally set different parameters for the software between generating the two tables. If the result is true, then discussion is warranted.

Figures 3, 4, 6, and 7. The authors offer the readers these four visually attractive figures with little explanation, either in the text or in the figure legends. As such, they should be deleted. Alternatively, there should be a clear explanation of the value of displaying them and what additional information is to be taken from viewing these figures that is more than the current statements of results in the text of the manuscript. The figure legends are insufficient. Tell the reader what the colors mean, what the connecting lines mean, what the distance between dots mean.

Authors do not indicate that they have placed their datasets into public databases, but they do state that they can be contacted to access the data.

6. PLOS authors have the option to publish the peer review history of their article (what does this mean?). If published, this will include your full peer review and any attached files.

Reviewer #1: No

Reviewer #2: No

---

## [Author Response · Author response to Decision Letter 0]

31 Mar 2021

Dear Editor Dr. Kogut,

We appreciate the comments from you and reviewers to improve this manuscript. We have made substantial changes to the Introduction and Discussion sections of the manuscript. We also made changes (red texts in the track-change version) according to the reviewers’ specific comments. We hope that with these updates the manuscript will fit well with Plos One’ standards. Below are the point-to-point responses to the reviewer’s comments. Thanks.

Sincerely,

Xiaolun Sun, PhD

Reviewer 1

Comment 1: Abstract – add the definition of SIC at the first time to mentioned it.

Response 1: We appreciate the reviewer’s suggestion and we have added the definition of the sub-inhibitory concentration (SIC) in the abstract (Lines 35-36).

Comment 2: Line 30: sub-inhibitory concentration; and Line 32: subinhibitory concentration – please, decide which one is correct the keep consistency.

Response 2: We have used “sub-inhibitory concentration” first and then “SIC” throughout the manuscript.

Comment 3: Introduction: Lines 62-64: the sentence does not make sense – please, re-write

Line 65: delete “its”

Line 71: “which contributes to intestinal inflammation”

Line 76 and others: “short-chain fatty acid”

Line 77: use GIT for gastrointestinal tract after the first time

Line 86: “..level, it remains..”

What is the hypothesis of the study?

The introduction needs a revision – sometimes the sentences are not connected and lack clarity.

Response 3: We are sorry for the lack of clarity. As per the reviewer’s suggestion we have substantial changes in the introduction with the hypothesis of our study (Line 55-95).

Comment 4: Discussion: Line 270: “colonize the intestine of the host”

Response 4: We have modified the sentence to “S. Enteritidis is an intracellular pathogen and induces an inflammatory immune response in GIT to overwhelm commensal microbiota, colonize and invade the intestinal cells (Fàbrega and Vila, 2013; Gart et al., 2016; Hallstrom and McCormick, 2011)” at Lines 268-270.

Reviewer 2

Comment 1: In the abstract, state what type of cells HTC are.

Response 1: We appreciate the reviewer’s suggestion, and we have added the description of HTC cells in the abstract (Line 33).

Comment 2: Because this experimental model uses a SIC of butyrate and has verified that the butyrate does not affect bacterial growth, the logical conclusion to be drawn about any detected protein differences between infected cells with and without added butyrate should be that these protein differences are not associated with bacterial growth. In the abstract and elsewhere, some of the stated conclusions seem to be in opposition to the data generated in this study, and instead are a reiteration of conclusions of previous studies reported in the literature, which are contradicted by the current study’s findings. The authors should state conclusions that are defensible from the data of their own study and, where needed, discuss the potential reasons for discrepancies in their own data and studies reported previously by others.

Response 2: We agree with the reviewer’s suggestion that SIC of sodium butyrate shouldn’t affect S. Enteritidis protein expression. We’re apologized for the lack of clarity. We have made changes to emphasize that the protein expression data in our study were macrophage proteins but not S. Enteritidis. 

Comment 3: Line 261. Should be reworded from “Following signaling pathway analysis by STRING revealed that…” to “Signaling pathway analysis by STRING predicted that…” The authors should be aware of the limitations of the analysis, and that it predicts but does not prove pathway involvement. And the word “Following” was not needed. 

Response 3: We have made changes the phrase in the lines 240 and 261 to “Signaling pathway analysis by STRING predicted that…”

Comment 4: Line 304 states that S. Enteritidis leads to apoptosis, but this study does not show a change in cell survival between infected and non-infected cells. Clarify. Do the authors mean that the amount of protein(s) typically associated with apoptosis are significantly different between infected and non-infected cells?

Response 4: We appreciate the reviewer’s comments and have modified the sentence to “Our results showed that S. Enteritidis infection upregulates proteins associated with ATP synthesis and cell apoptosis.” (Lines 298-299).

Comment 5: Line 336 states enhanced bacterial killing, but this experimental model showed no effect on bacterial growth.

Response 5: We appreciate the reviewer’s comment, and we modified the sentence to “It is necessary to investigate whether sodium butyrate-upregulated VIM protein activates inflammatory response.” (Lines 323-324)

Comment 6: Lines 342-343 state that butyrate upregulates proteins that enhanced bacterial killing in HTC, however, the experimental model used shows no effect of butyrate on bacterial growth. Thus, the logical conclusion to be drawn from their own data is that these upregulated proteins are not associated with bacterial growth and therefore, at least in HTC, are not enhancing bacterial killing.

Response 6: We appreciate the reviewer’s comments and we have modified the conclusion to “This study showed that butyrate reduced the cellular actin and cytoskeleton rearrangement proteins in S. Enteritidis infection in HTC cells. In addition, sodium butyrate upregulated proteins enhancing pro-inflammatory response in S. Enteritidis infected HTC cells. Collectively, these results suggest that sodium butyrate modulates HTC cell protein expression essential for S. Enteritidis invasion in the chicken macrophages.” (Lines 328-332)

Comment 7: Conclusions. Two of the major conclusions are about the effect of butyrate on “cellular invasion”, however, this study is not designed to, and does not, measure cellular invasion. The stated conclusions should be supported by the data of this study. The study has an appropriate design to detect effects of bacterial infection and of butyrate; the stated conclusions should related more closely to the data of this study.

Response 7: We appreciate the reviewer’s comments and we have modified the conclusion as showed in response 6. (Lines 328-332)

Comment 8: Tables 1 and 4. I cannot discern any organization of this table, because the proteins do not appear to be listed in order of any of the column headings. This makes it difficult to read. Please select a logical heading, for example the P value, and organize the table based upon it. That would add value for the reader.

Response 8: We appreciate the reviewer’s comment and we have made changes in the table (Tables 1 and 5) headings and incorporated two more tables (Tables 2 and 6) for more clarity.

Comment 9: Tables 2 and 3. Protein counts as low as 2 in pathways may not be highly reliable indicators of the involvement of that pathway (especially if it’s a pathway with many proteins) despite software predictions. Suggest that the authors consider more stringent thresholds for the pathways displayed so that the attention can focus on those with higher reliability. For example, an FDR of 0.02.

Response 9: We have used our previously published method to predict the pathway. We have modified the pathways analysis sentence to “To determine the signaling pathway of proteins, the differentially regulated proteins were analyzed using software such as Protein Analysis through Evolutionary Relationships software (PANTHER) and STRING protein association network (FDR 0.05) as described as before (Rath et al., 2019).”. (Lines 188-191)

Comment 10: Figure 1. This can be deleted and stated in a single sentence in the results section.

Response 10: We appreciate the reviewer’s comment and we have deleted Fig 1 from the manuscript and modified the statement in the results section to “Sodium butyrate at the SIC did not reduce the growth of S. Enteritidis compared to the control HTC cells (P>0.05)” (Lines 208-209)

Comment 11: Figure 2 legend. Please reword to state that “….biological processes were predicted..." by using STRING and PANTHER software.” The authors did not study the processes using software, they used the software to predict the involved biological processes. It’s important to recognize what the software does and does not do. It predicts (hypothesizes) the relationship of the proteins with processes.

Response 11: We appreciate the reviewer’s comment and we have reworded as, “Differentially expressed proteins were calculated using Scaffold software (P<0.05) and biological processes were predicted by using STRING and PANTHER software.” (Lines: 550-551 and 556-557) 

Comment 12: Figure 2 and 5. In comparing these two figures, it is apparent that Figure 2 allows the inclusion of the same proteins in multiple biological pathways because the results section reported 31 proteins as up- or down-regulated but Figure 2 displays well over 50 proteins in total count. However, in Figure 5, there is an exact agreement (n = 20) between the number reported in the results section and the number displayed in the graph. Although not impossible, this is highly unlikely to have such a discrepancy in the multiple use of proteins in Figure 2 and none in Figure 5. Authors should check that they have not accidentally set different parameters for the software between generating the two tables. If the result is true, then discussion is warranted.

Response 12: We appreciate the reviewer’s comment and we have deleted the figures 2 and 5. However, we have modified relevant sentences in Results section to “Quantitative comparison showed that S. Enteritidis infection downregulated 22 proteins and upregulated 9 proteins compared to uninfected HTC cells (P<0.05), however 358 proteins were not affected (P>0.05).” (Lines 215-217) and “HTC cells infected with S. Enteritidis in the presence of sodium butyrate downregulated 14 proteins and upregulated 6 proteins compared to HTC cells infected with S. Enteritidis alone (P<0.05), whereas 369 proteins were not affected (P>0.05).” (Lines 246-248).

Comment 13: Figures 3, 4, 6, and 7. The authors offer the readers these four visually attractive figures with little explanation, either in the text or in the figure legends. As such, they should be deleted. Alternatively, there should be a clear explanation of the value of displaying them and what additional information is to be taken from viewing these figures that is more than the current statements of results in the text of the manuscript. The figure legends are insufficient. Tell the reader what the colors mean, what the connecting lines mean, what the distance between dots mean.

Response 13: We are sorry for the lack of clarity. As per the reviewer’s suggestion we have deleted the figures 3, 4, 6, and 7 from our manuscript, but we added two more tables for better understanding the data (Tables 2 and 6).

---

## [Editor Report · Decision Letter 1]

5 Apr 2021

Sodium butyrate modulates chicken macrophage proteins essential for Salmonella Enteritidis invasion

PONE-D-20-40856R1

Dear Dr. Sun,

We’re pleased to inform you that your manuscript has been judged scientifically suitable for publication and will be formally accepted for publication once it meets all outstanding technical requirements.

Kind regards,

Michael H. Kogut, Ph.D.

Academic Editor

PLOS ONE
---

## [Editor Report · Acceptance letter]

19 Apr 2021

PONE-D-20-40856R1 

Sodium butyrate modulates chicken macrophage proteins essential for *Salmonella* Enteritidis invasion 

Dear Dr. Sun:

I'm pleased to inform you that your manuscript has been deemed suitable for publication in PLOS ONE. Congratulations! Your manuscript is now with our production department. 

Kind regards, 

on behalf of

Dr. Michael H. Kogut 

Academic Editor

PLOS ONE